# An Analysis of the Delayed Gradients Problem in Asynchronous SGD

**Ajay Jain**
Department of EECS
Massachusetts Institute of Technology
Cambridge, MA
`ajayjain@mit.edu`

**Anand Srinivasan**
AlphaSheets, Inc
San Francisco, CA
`anand@alphasheets.com`

**Parnian Barekatain**
IdeaFlow, Inc
Menlo Park, CA
`parni@ideapad.io`

## Abstract

Gradient descent can be effectively distributed across multiple workers via data parallelism or model parallelism. A common goal among all approaches is to minimize worker idle time. Parameter synchronization, such as after each minibatch in SGD, requires the parameter server to wait for the slowest worker to reply before applying an update. There is a fully asynchronous method (Dean et al., 2012), originally termed Downpour SGD, which minimizes worker idle time by allowing gradients computed on stale parameters to be sent to the parameter server. In practice, direct usage of asynchronous SGD leads to added noise during training from stale gradients (referred as the "delayed gradient problem"), which nontrivially decreases test accuracy. Delay compensation, such as that detailed by Zheng et al. (2016a), as well as various warm-start schemes, can help convergence. In this paper, we present a detailed analysis of the failure modes of ASGD due to delayed gradients under broad sweeps of hyperparameter selections. With a convolutional model, we find that learning rate and batch size selection are the majorizing factors in whether delayed gradients significantly reduce test accuracy. Careful selection of learning rate and batch size, or use of adaptive learning rate methods, is effective in minimizing the delayed gradient problem up to a large ($n = 257$) number of workers.

## 1 Introduction

Neural network training has scaled to many workers by both model parallelism, where a model is split across different workers, and data parallelism, where training data is sharded or distributed to worker copies. Distributed approaches differ in how model parameters are synchronized, but the fastest (gradients computed per unit of time) approaches are fully asynchronous and allow gradient updates computed from stale parameters (Dean et al., 2012).

Asynchronous optimization methods have advantages in shared clusters where individual workers may experience slow-downs from concurrent jobs, varying network conditions, or heterogeneous hardware. Chen et al. (2017) add additional backup workers such that a parameter server operating synchronous SGD can proceed to the next batch before the slowest workers reply.

However, due to the delayed application of gradients, ASGD suffers from accuracy degradation, for which various methods of delay compensation have been proposed:

- Dean et al. (2012) find that Adagrad adaptive learning rate optimization greatly increased the robustness of Downpour SGD.
- Chen et al. (2017) find that gradually introducing workers over the first 3 epochs is important for stability at high delay values. The authors also clip gradients for ASGD.

- Zheng et al. (2016b) adds the first-order term in the Taylor expansion of the gradient function to delayed gradient submissions.

To increase per-worker workload in synchronous training, Goyal et al. (2017) use large mini-batches (8k images) split across the cluster, and proportionally large learning rates. Even with ASGD, where mini-batches are not divided among workers, large mini-batches and learning rates would reduce the frequency of worker-server communications. We study the impact of batch size and learning rate on convergence in the presence of delayed gradients to better recommend exact use cases for ASGD.

## 2 METHODS

In our experiments, we train a LeNet-5 model for the MNIST digit recognition task over 30 epochs, using SGD with $m = 0.9$. To simulate delayed gradient submissions in asynchronous SGD, we create a wrapper around synchronous PyTorch optimizers which store gradients in a buffer and re-applies them after a constant delay.

To show that constant-delay synchronous SGD is a faithful simulation of gradient delay in fully asynchronous SGD, we conduct a Monte-Carlo simulation of N workers submitting gradients asynchronously to a parameter server. The time for a worker to process and submit a gradient is assumed to be normally distributed – that is, the time between gradient $t$ and $t'$ arriving at the server from worker $i$ is $T_{i,t'} - T_{i,t} \sim N(1, \sigma^2)$. Then, we find that the number of updates on the server before gradient $t'$ arrives fits a normal distribution centered at $N - 1$ (Figure 1, where $\sigma = 0.2$). While there is variance in this update count, the approximation of ASGD by a single worker with fixed simulated delay allows us to study the delayed gradient problem in isolation. Chen et al. (2017) conduct similar fixed simulations of delayed gradients on one worker in order to study the behavior of ASGD.

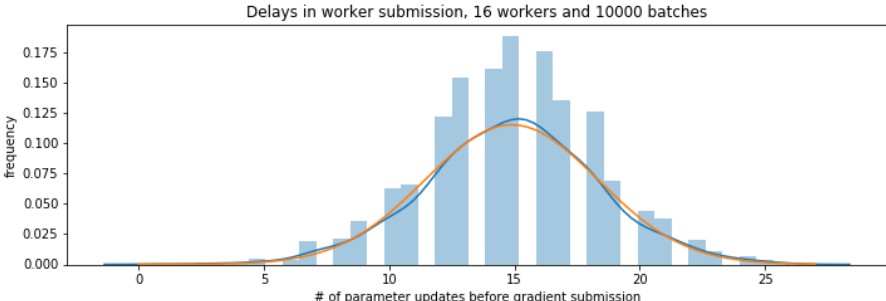

Figure 1: Distribution of update delay (in batches) for simulated ASGD. A normal fit is superimposed in orange.

Finally, to analyze the effect of delayed gradient application on test accuracy, we run the constant-delay SGD over a parameter sweep of learning rates, batch sizes, and delays. We also compare performance against Adam, which is expected to have lower sensitivity to initial learning rate.

## 3 EXPERIMENTAL RESULTS

Asynchronous SGD simulated at different submission delays shows best performance at low learning rates ($lr = 10^{-3}$) and low batch sizes ($b = 64, b = 128$). Test accuracy significantly degrades at batch sizes larger than 256 and $lr \geq 10^{-2.5}$. We observe that regardless of delay amount, test accuracy scales smoothly upward with $lr$, then sharply drops off. SGD with more delay is less resilient to learning rate adjustment, with the test accuracy dropping off sooner. One plausible explanation for low test accuracy at high learning rates in delayed SGD is that gradients are more variant near the start of training, so delayed updates add significant noise to the parameter search. Indeed, in Goyal et al. (2017), a gradual warmup schedule is used effectively to reduce test error for synchronous minibatch SGD with large minibatch sizes and proportionally scaled learning rates.

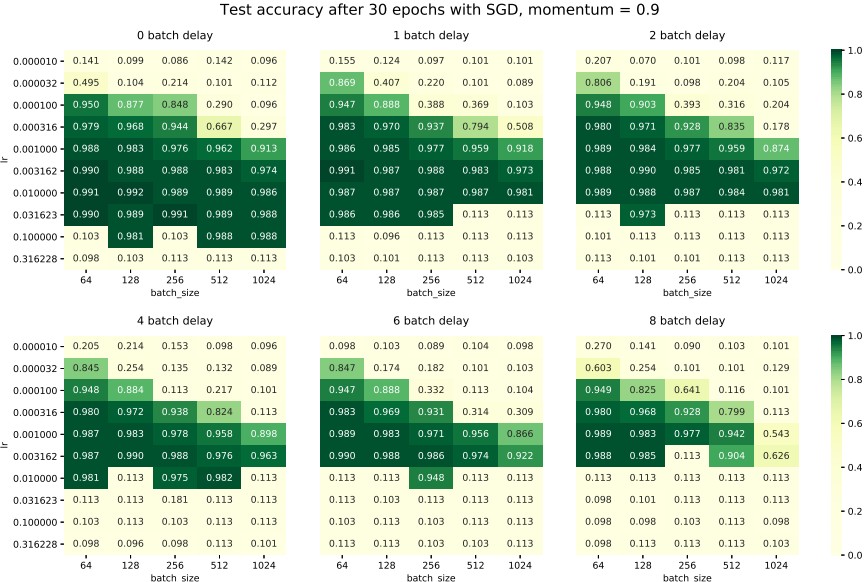

Figure 2: MNIST test accuracy under synchronous and delayed update schemes.

However, our experiments on Delayed SGD indicate that learning rate warmup only initially improves test accuracy but quickly falls off as soon as the full learning rate is applied. Learning rate schedule likely has little effect in mitigating delayed gradient noise. We find that optimizers which employ per-parameter adaptive learning rates, such as Adam (Kingma & Ba, 2014), increase resiliency to learning rate and batch size selections, achieving test accuracy comparable to baseline up to 256 delay, in comparison to SGD, which cannot handle more than 32 delay.

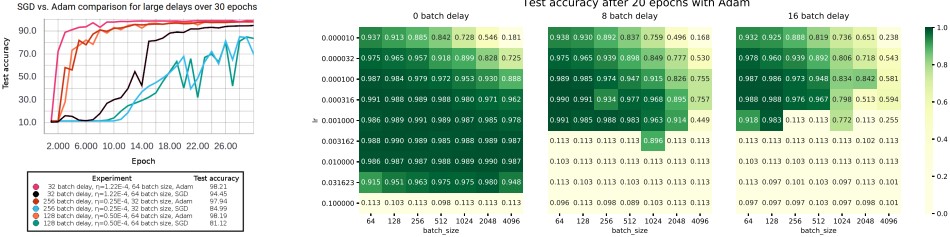

Figure 3: Using Adam instead of SGD, LeNet convergence is reached within 20 epochs even with delayed gradients for many batch sizes. Even so, there remains a sharp dropoff in accuracy at larger learning rates. On left, observe that SGD takes significantly longer to converge in the presence of substantial gradient delay.

Most importantly, we observe that, even for many-worker scenarios (delay=32, 128, and 256; Figure 3), hand-tuned learning rates and batch sizes using Adam can lead to results comparable (98.21%, 98.19%, and 97.94% accuracy respectively) to the 1-worker synchronous case. The heuristic we use to achieve this is roughly to halve the learning rate and halve the batch size when the number of workers is doubled. Figure (3) shows the dramatic difference in test accuracy convergence between SGD and Adam in these scenarios.

## 4 CONCLUSION

Our results highlight that learning rate and batch size are the major factors in ASGD stability. Furthermore, asynchronous gradient methods with adaptive optimizers and carefully chosen batch sizes can be a highly effective research tool for fast model search, even on large clusters.

ACKNOWLEDGMENTS

We would like to thank Jason Yosinski of Uber AI Labs for his advice throughout this research.

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
