# OpenReview forum: "An Analysis of the Delayed Gradients Problem in Asynchronous SGD"
_ICLR.cc/2018/Workshop — Reject_

### Official Review · AnonReviewer3 · 2018-03-09
**need deeper analysis on learning curves**

**Rating:** 3
**Confidence:** 4

**Review:**

The quality of writing needs big improvements. Most of claims made in Section 3 are made without reference to specific data points, making it very difficult to judge the validity of the claim. For example, authors mention that 'We observe that regardless of delay amount, test accuracy scales smoothly upward with lr, then sharply drops off', it is unclear which figure authors are referring to. None of the figures report results with different learning rate parameters. Another example: We find that optimizers which employ per-parameter adaptive learning rates, such as Adam (Kingma & Ba, 2014), increase resiliency to learning rate and batch size selections, achieving test accuracy comparable to baseline up to 256 delay, in comparison to SGD, which cannot handle more than 32 delay'. Again, the claim is made without reference to actual data point. In this case, I can guess that authors are referring to Figure 3, left. But it is unclear to me what range of learning rates were tried, and how authors chose specific learning rate and batch size each delay setting. The claim that 'SGD cannot handle more than 32 delay' should be refined to be more specific: When learning rates are very small, I believe SGD will be able to converge to a good solution, although it will take a long time.

Also, many of the claims authors make are applicable to synchronous SGD, and it is unclear how much of the effect they observe is due to the asynchronous aspect of optimization. Even in synchronous SGD, it is very well known that using smaller learning rates and smaller batch sizes is more stable than using larger learning rates and larger batch sizes.

---

### Official Review · AnonReviewer2 · 2018-03-10
**Useful to community, but not much novelty.**

**Rating:** 6
**Confidence:** 5

**Review:**

Summary:
The paper provides an analysis of the impact of delayed gradients in Asynchronous SGD. The authors create a simple model of delays and use that for the analysis. The focus is on impact of learning rate and batch sizes across two optimizers (ASGD and Adam). Adaptive optimizers do better than ASGD and are less sensitive to these two parameters. All experiments are performed for the LeNet-5 model on the MNIST dataset

Pros:
- Analysis shows the significant sensitivity to delayed gradients and the rapid reduction of the hyper-parameter space that yields good results  with increasing delays.
- This is consistent with what many papers over the years have shown but not analyzed earlier in this much detail.
- Adam is more resilient to delays and allows for a much larger range of learning rates and batch sizes even with longer delays

Cons
- Experiments are limited to 1 model on MNIST, it is unclear if they will hold for different kinds of models or larger datasets
- Not much novelty since the experimental results mostly agree with prior work

May be worth accepting primarily to share the details of the analysis with the community and to encourage more analysis in this area.

---

### Official Review · AnonReviewer1 · 2018-03-10
**Empirical evaluation of the performance of ASGD on various configurations of learning rate and batch size**

**Rating:** 4
**Confidence:** 4

**Review:**

The authors empirically evaluate the performance of ASGD on various configurations of learning rate and batch size. This suggests a guidance to the choice of these hyperparameters when using ASGD. Although it is useful practical information, I'm not sure suggesting a good hyperparameter range is enough contribution for publication. Also, the conclusion is drawn only from MNIST experiments. It is required to do more experiments on a broader range of tasks/datasets to make the claim more persuasive.

---

### Public Comment · ~Anand_Srinivasan1 · 2018-02-19
**Experimental code and data**

Code used to run experiments and data used to generate the plots in this paper can be accessed at https://github.com/ajayjain/outofsync-optim . There are instructions for getting started as well.

---

### Decision · Program_Chairs · 2018-03-20
**ICLR 2018 Workshop Acceptance Decision**

**Decision:**

Reject

**Comment:**

Based on the reviews, this paper has not been accepted for presentation at the ICLR workshop. However, the conversation and updates can continue to appear here on OpenReview.